# Behavioral and Physiological Responses of Therapy Dogs to Animal-Assisted Treatment in an Inpatient Stroke Rehabilitation Program

**DOI:** 10.3390/ani15020121

**Published:** 2025-01-07

**Authors:** Hao-Yu Shih, François Martin, Debra Ness, Whitney Romine, Taylor L. Peck, Tricia Turpin, Rachael Horoschak, Cindy Steeby, Hannah Phillips, Mary Claypool, Amanda Theuer, Grace M. Herbeck, Jasmine Sexton, Erin Pittman, Erica Bellamkonda, Nikita Maria Ligutam Mohabbat, Sandra A. Lyn, Brent A. Bauer, Arya B. Mohabbat

**Affiliations:** 1Division of General Internal Medicine, Mayo Clinic, Rochester, MN 55905, USA; bauer.brent@mayo.edu; 2Nestlé Purina Research, St. Louis, MO 63102, USA; francois.martin@rd.nestle.com (F.M.); patricia.turpin@rd.nestle.com (T.T.); rachael.horoschak@rd.nestle.com (R.H.); cynthia.steeby@rd.nestle.com (C.S.); hannah.phillips@rd.nestle.com (H.P.);; 3Department of Physical Medicine and Rehabilitation, Mayo Clinic, Rochester, MN 55905, USA; ness.debra@mayo.edu (D.N.); claypool.mary@mayo.edu (M.C.); theuer.amanda@mayo.edu (A.T.); bellamkonda.erica@mayo.edu (E.B.); mohabbat.nikita@mayo.edu (N.M.L.M.); 4Animal-Assisted Services, Mayo Clinic, Rochester, MN 55905, USA; romine.whitney@mayo.edu; 5Department of Medicine, Mayo Clinic, Rochester, MN 55905, USA; peck.taylor@mayo.edu (T.L.P.); herbeck.grace@mayo.edu (G.M.H.); sexton.jasmine@mayo.edu (J.S.); 6Volunteer Services, Mayo Clinic, Rochester, MN 55905, USA; pittman.erin@mayo.edu

**Keywords:** animal-assisted treatment/therapy, therapy dog, stroke, rehabilitation, salivary cortisol, salivary oxytocin, heart rate variability, tympanic membrane temperature, behavior, animal welfare

## Abstract

Therapy dogs have been increasingly incorporated into a variety of medical treatment programs, but research about the stress level of therapy dogs is limited. In this study, 14 therapy dog–handler pairs were embedded in an inpatient stroke rehabilitation program to provide animal-assisted treatment (AAT). Salivary endocrine, cardiac parameters, ear temperature, and behavioral observations were used to assess therapy dogs’ stress level during the AAT. The results suggested that incorporating AAT into an inpatient stroke rehabilitation program did not induce stress in the therapy dogs, and that the therapy dogs may have been more relaxed after the session.

## 1. Introduction

Dogs (*Canis familiaris*) have outstanding abilities for social interaction with humans, adapt well to various environments [1], and have been widely incorporated into animal-assisted treatment (AAT) [2,3,4]. The majority of studies designed to measure the efficacy of AAT focused specifically on human benefits, whereas how dogs are affected by participating in AAT activities is still not well understood. As a result, there is growing interest in understanding the effect of participating in AAT on dogs’ welfare. Many have, rightly so, opined that therapy dogs should not be negatively impacted by participating in AAT, and that they should enjoy it. As detailed in the 2018 IAHAIO white paper, “AAT should only be performed with the assistance of animals that are in good health, both physically and emotionally and that enjoy this type of activity. It is mandatory that handlers must be familiar with the individual animals taking part in an intervention. Professionals are held accountable for the well-being of the animals they are working with [5]”.

Glenk (2017) published an article regarding ethical and welfare concerns related to the use of dogs in animal-assisted interventions (AAIs), an umbrella term encompassing AAT in which animals are included for human benefits through different practices [3]. The author reviewed the limited body of knowledge on the impact of these activities on the welfare of therapy dogs. According to the review, researchers used multiple methods and techniques to evaluate the welfare of dogs (i.e., salivary, fecal, or hair cortisol, heart rate, questionnaires, interviews, behavior observation, cognitive tests). Some of the findings included outcomes usually associated with a negative welfare state (e.g., increased salivary cortisol levels, stress-related behaviors). However, in most instances, researchers reported no changes in the welfare indicators used or reported changes usually associated with a positive welfare state (e.g., increased play behavior, decreased hair cortisol concentration) [3]. Glenk (2017) concluded that, based on the data available, there were no acute concerns regarding dogs used in AAT [3]. Glenk and Foltin (2021) built on the 2017 work and reviewed an additional 11 peer-reviewed studies published between 2017 and 2021 that assessed the welfare of dogs participating in AAI [2]. The methods and techniques used to measure the welfare of the therapy dogs were similar to those reported in 2017, with the addition of salivary oxytocin, tympanic ear temperature, and respiration rate. The results reported in the 11 studies suggested that participating in AAI activities had a negligible negative impact on dog wellbeing. Some of the welfare indicators did show significant changes, but these changes remained within normal physiological ranges. Furthermore, these changes could have reflected changes in arousal states in the dogs, and not necessarily a sign that their welfare was compromised [2].

Reviewing the literature for peer-reviewed scientific papers published after the Glenk and Foltin review (2021) led to the identification of three additional publications [2]. Friedmann et al. (2022) measured stress via heart rate variability (HRV) in 25 shelter dogs involved in a dog walking program for veterans [6]. Pre-walk HRV values were not different from walk/post-walk HRV values. The authors concluded that the shelter dog walking program did not induce stress in the participating dogs [6]. Kohoutkova et al. (2024) measured salivary cortisol in 15 dogs participating in AAI activities with children and seniors [7]. The authors asked the handlers/owners to complete a questionnaire about the behavior of their dog during the AAI activities. There were no differences in cortisol between AAI days vs. non-AAI days. Some stress-related behaviors were reported during AAI sessions, but no comparisons with non-AAI days were provided [7]. Based on these findings, it appeared that AAI sessions did not induce significant stress in the dogs. In a study about the efficacy of canine socialization for veteran students, Webberson et al. (2024) evaluated the wellbeing of 30 participating shelter dogs using behavior as their welfare indicator [8]. The authors reported more fear-related behavior at the beginning of the sessions, compared to mid- and end of sessions. They also observed a decrease in overall behavioral activity at the end of the session, compared to the beginning of the session. They concluded that AAI did not induce stress in dogs, and dogs may have been calmer at the end of the session [8].

In previous work, to gain an in-depth understanding of how participating in AAT could impact the wellbeing of therapy dogs, the present authors proposed a methodology composed of various noninvasive physiological parameters [4]. The physiological response of 19 therapy dogs interacting with patients with fibromyalgia was measured. Results showed that dogs had a reduced heart rate and lower right tympanic membrane temperature (TMT) after the visit, along with all other emotional parameters (i.e., heart rate variability (HRV), salivary cortisol and oxytocin concentrations) remaining stable. It was concluded that the therapy dogs had not been affected negatively by their participation in AAT, and they may have been in a more relaxed state after visiting the patients [4].

This study is the first randomized–controlled and prospective study that investigated the wellbeing of therapy dogs in an inpatient stroke rehabilitation program. A similar multimodal approach, involving both physiological and behavioral parameters [4], was utilized to measure the welfare of AAT dogs working with a different clinical population and directly involved in the patients’ treatment program. The physiological parameters included salivary oxytocin, salivary cortisol, tympanic membrane temperatures, and cardiac activity (HR, HRV). Furthermore, the Canine Fear, Anxiety, and Stress (FAS) scale was applied, and an in situ behavior evaluation was performed. This scale involves observing canine behavior and body language and has been clinically and scientifically utilized to assess the stress level of dogs [9,10,11]. The working hypothesis was that, based on the behavioral and physiological results, the wellbeing of therapy dogs would not be negatively impacted by their participation in stroke rehabilitation AAT activities.

## 2. Materials and Methods

This study was reviewed and approved by Mayo Clinic’s research committee (IRB #: 21-005175; Protocol: A00006122-21).

### 2.1. Therapy Dogs and Handlers

Fourteen therapy dog–handler pairs that were part of the Mayo Clinic’s Caring Canine Program [12] participated in this study. All dog–handler pairs passed an evaluation that assessed the dog’s responses to the environment and their interaction with humans, including people of different ages, mobilities, and interactive styles. The handler’s involvement during the exercise was also evaluated. All dog–handler pairs were registered as therapy dogs with Alliance of Therapy Dogs [13] or Pet Partners [14]. All therapy dogs were at least 1 year of age, up to date on vaccines, deemed healthy by their veterinarian, and were not fed a raw diet. Of all the therapy dogs, nine were female (eight spayed and one intact) and five were male (four neutered and one intact). The dogs were always accompanied by their owners/handlers (eleven women and three men) (Table 1). All dog–handler pairs volunteered at Mayo Clinic on a regular basis and voluntarily participated in this study. All handlers abided by the Mayo Clinic Volunteer and Caring Canine program [12].

### 2.2. Human Participants

Fifty patients were recruited and randomly assigned to either the experimental group (n = 25, standard stroke rehabilitation plus AAT) or control group (n = 25, standard stroke rehabilitation). All study participants were patients with a recent stroke who were hospitalized in the Stroke Rehabilitation Unit of Mayo Clinic, Saint Marys Campus (Rochester, MN), and enrolled in the inpatient brain rehabilitation program between February 2023 and February 2024. To be enrolled in this study, all participants had to (1) be 18 years of age or older; (2) be enrolled in the comprehensive inpatient stroke rehabilitation program; (3) have had a recent stroke from any cause; (4) be able and willing to give informed consent; and (5) be able to speak English. Patients with any of the following conditions were excluded: (1) allergic to or fearful of dogs; (2) history of pacemaker implantation; (3) pregnant; or (4) deemed inappropriate for this study by the medical or study teams.

### 2.3. General Methods

All AAT sessions were conducted using the diamond model, involving patients, therapy dogs, therapy dog handlers, and therapists. All AAT sessions and data were conducted and collected in the afternoons between 1 pm and 4 pm. Prior to the AAT session, in the sample collection room separate from the patient and away from distractions, the researcher (HYS) collected the dog’s saliva, simultaneous bilateral tympanic ear temperature, and fitted the HRV sensor around the dog’s chest. The HRV recording continued for 5 min while the dog and the handler remained in the sample collection room before the AAT session started. This gave the dog time to settle prior to the session and captured its cardiac activity at rest. The total pre-session procedure took approximately 15 min. After this, HYS led the handler–therapy dog pair to the patient’s room; on the way, the handler–dog pair was instructed to avoid any interactions with others. Before meeting the patient, outside of the patient’s room, the dog’s behavior was rated using the Canine FAS scale. The handler–therapy dog pair then entered the patient’s room and had a brief greeting with the patient and the rehabilitation therapist. The patient was allowed to interact with the therapy dog and the dog had the chance to become familiar with the patient and the therapist. The therapist would then also introduce the rehabilitation plan to the patient and therapy dog handler.

Following the introduction, the whole team headed to the inpatient rehabilitation therapy treatment space. During the stroke rehabilitation session, the therapy dog–handler pair were actively involved and integrated into the occupational and physical therapy sessions for mobility, fine motor coordination, visual scanning, cognitive tasks, etc. The session lasted for 30 min. HYS was present throughout the therapy session but did not intervene. HYS is a veterinarian and, while it was not necessary during this study, was prepared to intervene or end an AAT rehabilitation session should a dog show significant signs of stress or discomfort. Exercises and activities performed during the session were determined by the therapist, patient, therapy dog handler, and therapy dog. Details about the exercises and activities performed during the session will be introduced in another paper focusing on the effects of the AAT-embedded stroke rehabilitation program on the human patients.

After the session, the handler, the dog, and HYS escorted the patient’s back to their room. HYS rated the dog’s behavior post-interaction using the Canine FAS scale right after leaving the patient’s room. The dog and the handler returned to the sample collection room by themselves and were asked to stay there and rest for about five minutes. The researcher then returned and collected saliva and bilateral tympanic ear temperature of the dog and removed the HRV sensor from the dog. The entire post-session procedure took around 15 min.

The sample collection room, treatment space and all patient rooms were on the same floor within 2 min walking distance of each other. The treatment space included a physical therapy gym, an occupational therapy gym, and a rehabilitation simulation apartment. Equipment in the physical therapy gym included, but was not limited to, gait trainers, stairs, an exoskeleton, weights, a treadmill, balance tools, parallel bars, and exercise bikes; equipment in the occupational gym included, but was not limited to, therapy mats, fine motor coordination tools, and a reaction trainer. The rehabilitation simulation apartment was to replicate a real-life home living environment, which consisted of a bedroom, kitchen, and bathroom. In addition to the treatment space, some portions of the AAT sessions took place in the hallway on the same hospital floor, just outside of the treatment space, or in the garden, depending on the specific treatment plan. The garden was on the first floor outside of the building and away from other human interactions and distractions.

All handler–therapy dog pairs were familiar with the treatment space/patient rooms either because they had visited patients there before or were given an opportunity to familiarize themselves with the environment prior to the study. To avoid the therapy dog being distracted, only one handler–therapy dog pair was present at a time, and people other than the patient and the research team were instructed not to interact with the dog. Therapy dogs D2 and D3 sometimes showed up at the same time, but they were instructed not to be present in the same space during the research period. Prior to the start of this study, approximately seventy therapists participated in an introductory lecture that included an overview of the research study, dog-assisted treatment, and therapy dog welfare. According to the Mayo Clinic Caring Canines Policy, a therapy dog should work no more than two hours per day. In this study, dogs participated in a 30 min session, with the entire process lasting approximately an hour for each visit, including pre-session and post-session procedures. Handler–dog pairs were randomly assigned to individuals, and they did not know the patient prior to the visit.

### 2.4. Physiological Measurements and Behavior Evaluation

#### 2.4.1. Cardiac Activity

Canine cardiac activity was monitored continuously throughout the therapy session using Polar V800 and Polar Pacer Pro heart rate monitor (Chicago, IL, USA), which included a receiver (watch) and a sensor-transmitter (soft elastic belt with electrodes). The belt was fitted onto the dog’s chest and a water-based electrode lubricant was applied on the belt to ensure good contact with the chest. The analysis window of HRV was one minute pre- and post-session because the data collection time was very limited due to patients’ rehabilitation/treatment schedule and hospital logistics. However, time-domain metrics (e.g., RMSSD, SDNN) have been shown to be reliable indicators of heart rate variability when using short recording times [15]. After the session, the data were downloaded using the Polar Flow application (Polar Electro Öy, Kempele, Finland). The data were analyzed using Kubios HRV Scientific software (version 4.0.3) (Kubios Öy, Kupio, Finland) [16]. Cardiac parameters were compared pre- and post-session while the dog was at rest. The cleanest one-minute intervals pre- and post-session were selected to provide standardized time points for comparison (artifact correction of less than 5%). Measured cardiac parameters included heart rate (HR), the time between two successive R-waves (RR interval), the root square mean of the successive differences in RR intervals (RMSSD), standard deviation of normal RR intervals (SDNN), and the ratio of RMSSD to SDNN (RMSSD:SDNN ratio). Decreased HRV (RR interval, RMSSD, SDNN, and RMSSD:SDNN ratio) values and increased HR are associated with increased stress and activity levels.

#### 2.4.2. Salivary Cortisol and Oxytocin

Saliva was collected before and after the therapy session using the Super•SAL™2 (Oasis Diagnostic LLC, Vancouver, WA, USA) saliva collection swab. The swab was held in the dog’s mouth until saturated over the course of two minutes and the dog was allowed to chew the swab. Samples were immediately placed in a cooler filled with ice packs after collection. Samples were prepared for analysis by cutting the sponge part of the swab into a 3 mL syringe and extracting the saliva into a 1.5 mL Eppendorf by depressing the plunger. The Eppendorf was then centrifuged at 2000× *g* for two minutes, and the supernatant was transferred to another labeled Eppendorf and was stored in a −80 °C freezer until being shipped on dry ice to Nestlé Purina’s laboratory for cortisol and oxytocin analyses.

Salivary cortisol was measured by Salimetrics Expanded Range High Sensitivity Salivary Cortisol Enzyme Immunoassay Kit (Salimetrics, State College, PA, USA). On the day of analysis, frozen saliva samples were thawed completely, vortexed, and centrifuged at 1500× *g* for 15 min. Clear saliva supernatant was pipetted, in duplicate, into the appropriate wells of the plate and processed according to kit instructions. Plate was read at 450 nm in a plate reader within ten minutes of stopping the reaction.

Salivary oxytocin was measured by a liquid chromatography–mass spectrometry (LCMS) platform containing a Nexera X2 ultra-high performance liquid chromatograph (Shimadzu, Columbia, MD, USA) and an AB Sciex Triple Quad 6500+ mass spectrometer (AB Sciex, Framingham, MA, USA). A total of 150 µL of saliva sample was diluted with 600 µL 80% aqueous acetonitrile (ACN) containing 1 nM deuterated oxytocin as the internal standard. After vortexing for 30 s, the samples were centrifuged at 15,000× *g* at 4 °C for ten minutes. The supernatant was transferred to a 96-well plate of 1.2 mL strip microtubes and evaporated under continuous nitrogen until dry with a Microvap microplate evaporator (Organomation, West Berlin, MA, USA). The dried samples were reconstituted with 50 µL of 50% aqueous ACN, vortexed for 30 s, and then transferred to HPLC vials for LCMS analysis. Chromatograms and mass spectral data were processed and quantified using Analyst 1.7.3 and MultiQuant 3.0.3 software (AB Sciex).

Increased salivary cortisol concentrations are associated with higher stress levels, while elevated oxytocin concentrations are linked to reduced stress.

#### 2.4.3. Tympanic Membrane Temperature

Bilateral tympanic membrane temperature was taken prior to and after the rehabilitation session using a tympanic thermometer (Welch Allyn Thermoscan Pro 4000, Welch Allyn, Inc., Skaneateles Falls, NY, USA). The temperature of both ears was taken simultaneously [4,17]. A warmer right tympanic temperature is generally believed to be related to increased stress level.

#### 2.4.4. Dog Behavior Evaluation

The Canine Fear, Anxiety, and Stress (FAS) scale is a tool that has been utilized in veterinary clinics and research to assess the stress level in dogs. Based on the observation of behavior and body language (from panting, ears pinned back, and tail tucked to barking, growling, and snapping), this Likert-type scale (relaxed/severe stress) provides a global FAS score for each dog. A higher FAS score indicates that the dog is showing more pronounced signs of fear, anxiety, and stress. For instance, a dog with a FAS score 0 is relaxed with loose facial muscles and body tension, whereas a dog scored 5 on the FAS scale is severely fearful, anxious, and stressed, showing signs of tense facial and body tension and offensive or defensive aggression [9,10,11]. The FAS scale was administered immediately before and after the therapy session outside of the patient’s room.

### 2.5. Statistical Analysis

Descriptive analyses were used to summarize the demographic details of patients. All outcomes were analyzed by linear mixed models in RStudio (version 2024.04.02) using the lme function from the nlme package (v.3.1-164). The fixed effect was “session” (2 levels; pre-session and post-session). The random effect was “dog” (14 levels), which accounted for repeated measures within dog. All outcomes were analyzed on the natural scale except cortisol and oxytocin, which were analyzed after adding a constant of 1 then log transforming. Transformations were applied to normalize residuals and improve model fit. Results for outcomes analyzed on transformed scales were back-transformed to the natural scale.

## 3. Results

Detailed results are presented in Table 2.

### 3.1. Cardiac Activity

Compared to pre-session, therapy dogs showed greater parasympathetic dominant responses in various cardiac parameters, with a lower mean HR and higher HRV indicators. These differences were statistically significant. The mean HR decreased from 96.97 beats/min pre-session to 79.62 beats/min post-session (F = 74.6, *p* < 0.0001). The RR interval went from 640.96 to 798.78 (F = 65.3, *p* < 0.0001); RMSSD went from 133.48 to 253.30 (F = 49.4, *p* < 0.0001); SDNN increased from 100.14 to 175.76 (F = 56.4, *p* < 0.0001); and the RMSSD:SDNN ratio went from 1.24 to 1.37 (F = 13.07, *p* = 0.0005).

### 3.2. Salivary Cortisol and Oxytocin

Salivary cortisol and oxytocin levels were numerically lower post-session when compared to levels pre-session. However, the differences were not statistically significant. The salivary cortisol concentration decreased from 0.29 (μg/dL) pre-session to 0.27 (μg/dL) post-session (F = 2.34, *p* = 0.13), and the salivary oxytocin concentration went from 1.76 (nM) pre-session to 1.47 (nM) post-session (F = 1.96, *p* = 0.17).

### 3.3. Tympanic Membrane Temperature

The right tympanic membrane temperature increased from 38.18 °C pre-session to 38.27 °C post-session (F = 4.58, *p* = 0.034). This increase was statistically significant. There was no significant difference in left tympanic membrane temperature before (38.17 °C) and after the session (38.20 °C).

### 3.4. Dog Behavior Evaluation

The Canine FAS scores were similar pre- (0.48) and post-session (0.46).

## 4. Discussion

Using behavioral and noninvasive physiological measures, this study investigated the welfare of therapy dogs incorporated into 30 min AAT sessions in an inpatient stroke rehabilitation program between February 2023 and February 2024. From the battery of behavioral and physiological measures collected, the overall findings indicate that certified therapy dogs were not adversely affected by their participation, and that they may have even been more relaxed following the AAT session.

HR and HRV are validated objective measures for evaluating canine stress response through detecting sympathetic and parasympathetic nervous system activity [18]. Decreased HRV values and increased HR are associated with increased stress and activity levels (sympathetic dominance); by contrast, increased HRV values and decreased HR are related to lower stress and activity levels (parasympathetic dominance) [18,19,20]. In dogs, a higher HR and decreased HRV values have been associated with stress-related behaviors [20]. In the present study, therapy dogs had a decreased HR and increased HRV values after participating in rehabilitation sessions, indicating parasympathetic dominance and thus lowered physiological stress states.

In this study, the therapy dogs’ cortisol and oxytocin levels remained stable pre- and post-session. These results corroborate the present team’s earlier work on the wellbeing of therapy dogs [4]. However, as previously reported in other studies, variable effects of working sessions on therapy dogs’ endocrine activity leave matters open to further research [2]. It may be that differences observed between studies were due to research artifacts. Differences in cortisol matrices (i.e., fecal, hair, saliva) and research designs may explain these discrepancies better than actual biological differences. Standardization of how the welfare of therapy animals is evaluated may help minimize these differences and improve our understanding of the true effects of AAT programs on therapy dogs.

In the human literature, warmer right tympanic temperature is generally believed to be related to stress [21]; similarly, a feline study showed that lower right tympanic temperature was associated with lower cortisol concentration (lower stress/arousal level) [22]. In canines, it was reported that decreased temperature in both ears was associated with isolation stress [23], while increased bilateral ear temperature was related to arousal [24]; yet no significant temperature difference was observed between ears in either study. Another study found that increased TMT asymmetry, regardless of directionality, was related to enhanced attentiveness [25]. In the authors’ previous study, therapy dogs had lower right TMT after the AAT sessions and were more relaxed [4]. In the current study, however, the therapy dogs’ right TMT was higher post-session, which may suggest an increase in stress. However, when the results of all the welfare indicators are considered altogether (i.e., decreased HR and increased HRV, unchanged cortisol and oxytocin levels, similar behavior presentation pre- and post-session), a different conclusion emerges: the welfare of the dogs in this study was not negatively impacted by their participation in AAT activities. The increased right TMT post-session observed may be explained by the fact that the therapy dogs in this study were actively involved in AAT, meaning they participated in exercises alongside the patients, rather than a passive involvement where the dogs simply provide companionship without engaging in specific activities. The increase in right TMT might reflect increased attentiveness [25] during the AAT, as dogs were more active and asked to execute specific tasks while interacting with patients and therapists. Additional research is needed to fully understand how variations in activity levels/attentiveness during AAT sessions impact TMT in therapy dogs.

This study has shortcomings that limit the generalizability of its conclusions. It was conducted in a single medical center with one patient population, and patients were predominantly elderly (66 ± 11.5 years old) and Caucasian (80%). The dog–handler pairs participated in only one relatively short AAT session per day, and usually not more than once a week. All dog–handler pairs possessed high skill and training levels (they were all certified by a national AAT organization and were also enrolled in the Mayo Clinic’s Caring Canine Program). These factors may not be representative for most therapy dogs. However, they would suggest at least minimum best practices to support the wellbeing of dogs in such programs.

In this study, the authors used a rapid behavior evaluation tool (Canine FAS) and only one person administered it. Quantifying the canine behavioral stress response using video recording, detailed ethograms, and inter-rater reliability is a more objective and robust approach [26], but due to patients’ privacy, we were not able to record the rehabilitation sessions. However, because we used multiple noninvasive physiological indicators of stress that provide solid evidentiary support for the positive welfare status of the dogs, we are confident in the evaluation of the dogs’ behavior.

Finally, while the pre- and post-session results indicated that the careful inclusion of therapy dogs in an inpatient stroke rehabilitation program causes minimal to no stress to the animals, it is important to remain mindful that each visit could still be a potential source of stress. Haubenhofer et al. [27] reported that therapy dogs exhibited higher salivary cortisol concentrations on workdays compared to non-workdays. However, further investigation is needed to determine whether the elevated cortisol levels are directly related to stress. Future research should account for therapy dogs’ baseline stress levels in the study design to provide more comprehensive insights.

## 5. Conclusions

This study increases the overall knowledge of the welfare of therapy dogs. This is the first randomized–controlled and prospective study that investigated the wellbeing of therapy dogs in an inpatient stroke rehabilitation program using multiple noninvasive physiological indicators and behavior observation. This study was conducted in a real-world hospital setting, enhancing its generalizability to other clinical environments. The behavioral and physiological responses of therapy dogs indicate that the careful incorporation of therapy dogs into an inpatient stroke rehabilitation program causes little to no stress to the animals, even when they are asked to be physically active during the sessions.

## Figures and Tables

**Table 1 animals-15-00121-t001:** Information about therapy dog–handler pairs.

Dog	Breed	Age	Dog’s Sex	Handler’s Gender	Dog’s Weight (kg)	Therapy Dog Certification Organizations	Experience as a Therapy Dog	Dog’s Past Experience with Patients with a Stroke	Numbers of Sessions Completed During This Study
D1	Old English Sheepdog × Standard Poodle	5 yr	FS	M	29	Pet Partners	1 year	None	13
D2	Maltese × Yorkshire Terrier	9 yr	FS	F	5	Alliance of Therapy Dogs	5 years	Seen 10–15 patients with a stroke	5
D3	Maltese × Yorkshire Terrier	9 yr	FS	M	5	Alliance of Therapy Dogs	5 years	Seen 10–15 patients with a stroke	12
D4	Labrador Retriever × Golden Retriever	7 yr	MN	F	23.6	Alliance of Therapy Dogs	4 years	Some experience with patient visits	13
D5	Cardigan Welsh Corgi	6 yr	MI	F	17.7	Alliance of Therapy Dogs	3.5 years	Seen > 15 patients with a stroke	2
D6	Australian Shepherd	9 yr	FS	F	16.8	Pet Partners	1 year	None	5
D7	Australian Shepherd	6 yr	MN	F	21.3	Pet Partners	4 years	Seen < 10 patients with a stroke	2
D8	Corgi	3 yr	FS	F	19.1	Alliance of Therapy Dogs	6 months	Had a family member who had a stroke	1
D9	Yorkshire Terrier × Toy Poodle	12 yr	FS	F	1.8	Alliance of Therapy Dogs	5 years	None	1
D10	Labrador Retriever	6 yr	FS	M	35.4	Pet Partners	5 years	None	8
D11	American Eskimo Dog	6 yrs	FI	F	14.5	Alliance of Therapy Dogs	9 months	None	1
D12	English Springer Spaniel	5 yrs	MN	F	31.8	Pet Partners	9 months	None	6
D13	Chihuahua mix	4 yrs	MN	F	6.8	Pet Partners	1 year	Once	3
D14	Australian Shepherd	3 yrs	FS	F	22.7	Pet Partners	4 months	None	2

D2 and D3 were littermates. F (female); M (male); FS (female spayed); FI (female intact); MN (male neutered); and MI (male intact).

**Table 2 animals-15-00121-t002:** Physiological and behavioral parameters of therapy dogs pre- and post-therapy session.

Parameter		Pre	Post	Post-Pre
		μ (SE or CI) n	μ (SE or CI) n	*p* Value of the F-Test
**Heart rate (beats/min)**		96.97 (4.91) n = 12	79.62 (4.88) n = 13	** *p < 0.0001* **
**RR interval (ms)**		640.96 (43.10) n = 12	798.78 (42.82) n = 13	** *p < 0.0001* **
**RMSSD**		133.48 (31.19) n = 12	253.30 (30.92) n = 13	** *p < 0.0001* **
**SDNN**		100.14 (17.46) n = 12	175.76 (17.31) n = 13	** *p < 0.0001* **
**RMSSD:SDNN ratio**		1.24 (0.067) n = 12	1.37 (0.066) n = 13	** *p = 0.0005* **
**Salivary cortisol (μg/dL)**		0.29 (0.21, 0.39) n = 18	0.27 (0.18, 0.36) n = 18	*p* = 0.13
**Salivary oxytocin (nM)**		1.76 (1.08, 2.65) n = 13	1.47 (0.84, 2.30) n = 13	*p* = 0.17
**Tympanic membrane temperature (°C)**	Left	38.17 (0.10) n = 14	38.20 (0.10) n = 13	*p* = 0.50
	Right	38.18 (0.11) n = 13	38.27 (0.11) n = 13	***p* = 0.034**
**Canine FAS level**		0.48 (0.17)	0.46 (0.17)	*p* = 0.86

## Data Availability

The data presented in this study are available on request from the corresponding author due to privacy issues.

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
