# Peer review of "Behavioral and Physiological Responses of Therapy Dogs to Animal-Assisted Treatment in an Inpatient Stroke Rehabilitation Program"

_animals, 2025, doi:10.3390/ani15020121_

Round 1
Reviewer 1 Report
Comments and Suggestions for Authors
Very interesting research topic. Congratulations to the Authors for the idea and execution.
Introducing animals to human clinics/hospitals is usually a big challenge.
I have a few questions:
- why was information provided about the division of patients into two groups with AAT and without AAT? Why was there a control group without AAT?
- why is there such a difference in the number of sessions (from 1 to 13) in individual dogs?
- was the age of the dog taken into account? The age range is large from 3 to 12 years (body temperature, heart rate are different in young and old individuals)
- why was the influence of the sex of the handler on stress parameters in dogs not checked?
- why were the results not described by sex of the dogs?
- who assessed the behavior (FAS) of the dogs? a behaviorist, zoopsychologist, veterinarian?
- why was it not checked whether there are correlations between individual parameters, e.g. cortisol level with body temperature, HR and FAS index?
- additionally, if I understand correctly, the heart rate measurement was only for a minute before and a minute after the rehabilitation session?
Was the electrode band put on and taken off each time?
In my opinion, the conclusions are too general and too cautious.
Author Response
|
Very interesting research topic. Congratulations to the Authors for the idea and execution. Introducing animals to human clinics/hospitals is usually a big challenge. |
Thank you very much for reviewing this manuscript. Below, please find our responses to your comments. |
|
Why was information provided about the division of patients into two groups with AAT and without AAT? Why was there a control group without AAT? |
This is a randomized-controlled study to investigate the effect of AAT. Therefore, patients were randomly assigned into either the control group (without AAT) and the experimental group (with AAT). |
|
Why is there such a difference in the number of sessions (from 1 to 13) in individual dogs? |
These therapy dog-handler pairs were volunteers, so the number of sessions attended was determined by their availabilities. |
|
Was the age of the dog taken into account? The age range is large from 3 to 12 years (body temperature, heart rate are different in young and old individuals) |
The age of the dog was not taken into account. Yes, the age is correlated with different physiological parameters. Therefore, we compared all parameters before and after each session of the SAME dog (paired/within individual comparison). It would be interesting to investigate the effect of age on these factors by including a larger sample size in the future. |
|
Why was the influence of the sex of the handler on stress parameters in dogs not checked? |
The aim of this study was to test the effect of AAT using a paired/within-comparison approach. The influence of sex was unable to be examined due to the research design and the sample size. It would be interesting to investigate the effect of sex on canine stress by including a larger sample size in the future.
For your reference, one of our authors (HYS) published a relevant study about the effect of human gender and canine sex on their behavioral interaction, including stress response. See below for information for the paper. “Who Is Pulling the Leash? Effects of Human Gender and Dog Sex on Human–Dog Dyads When Walking On-Leash” https://doi.org/10.3390/ani10101894 |
|
Why were the results not described by sex of the dogs? |
As mentioned earlier, the aim of this study was to test the effect of AAT using a paired/within-comparison approach. The influence of sex was unable to be examined due to the research design and the sample size. It would be interesting to investigate the effect of sex on canine stress by including a larger sample size in the future.
Again, please see the paper below published by one of our authors (HYS) about the effect of human gender and canine sex on human-dog interaction, including stress response. “Who Is Pulling the Leash? Effects of Human Gender and Dog Sex on Human–Dog Dyads When Walking On-Leash” https://doi.org/10.3390/ani10101894 |
|
Who assessed the behavior (FAS) of the dogs? a behaviorist, zoopsychologist, veterinarian? |
The behavior (FAS) was assessed by the first author, HYS. HYS is a Fear Free certified veterinarian and a certified dog trainer. He completed his PhD in human-dog behavioral interaction that involved assessing behaviors of over 200 shelter dogs and over 4,000 minutes of canine behavior footages using ethograms. (Here is his publication profile: https://orcid.org/0000-0002-5630-8632) |
|
Why was it not checked whether there are correlations between individual parameters, e.g. cortisol level with body temperature, HR and FAS index? |
The aim of this study was to test the effect of AAT using a paired/within-comparison approach. The correlations among individual parameters were unable to be examined due to the research design and the sample size. It would be interesting to investigate the correlation by including a larger sample size in the future. |
|
Additionally, if I understand correctly, the heart rate measurement was only for a minute before and a minute after the rehabilitation session? |
The heart rate monitor was fitted on the dog the all-time during the therapy until removal. However, due to the research design, logistic and data quality, only one minute of data before and after therapy respectively was used for analysis. |
|
Was the electrode band put on and taken off each time? |
The heart rate monitor/electrode band was fitted on the dog the all-time during the therapy until removal. However, due to the research design, logistic and data quality, only one minute of data before and after therapy respectively was used for analysis. |
|
In my opinion, the conclusions are too general and too cautious. |
Thank you for pointing this out. We’ve added that behavioral and physiological findings specifically suggested that therapy dogs experienced minimal stress during the process. |
Reviewer 2 Report
Comments and Suggestions for Authors
This study examined responses of therapy dogs to participation in an animal-assisted rehabilitation program for stroke patients. Strengths of this study include a nice sample size for this type of work (14 pairs of therapy dogs and their handlers), multiple physiological measures plus behavioral measures used to assess welfare, and its design as a prospective randomized-controlled study. One weakness was failure to include the Canine Fear, Anxiety, and Stress scale as an appendix and/or provide more descriptive information about this scale so that readers could put the behavioral results in a larger context. Most of my comments are editorial in nature and are detailed below.
Abstract:
If possible, I would include that the study was prospective in nature and randomized-controlled somewhere early in the Abstract.
Introduction:
Lines 57-58: Given the numerous terms used when describing human - nonhuman animal interactions, it might be helpful to differentiate AAT and AAI, mentioning the latter is a broader term that encompasses AAT as well as other animal-assisted activities.
Materials and Methods:
Line 129: Why is it important to mention that the dogs were not fed a raw diet?
Table 1: Consider using the following abbreviations for dog sex and spay/neuter status: SF (spayed female), IF (intact female), NM (neutered male), and IM (intact male) and F and M for female and male handlers, respectively. This might improve column spacing and make the table more readable. These could be described in a footnote. Note that there is a footnote numbered 2 after the table stating D2 and D3 were littermates, but there is no 2 within the table.
Line 207: I suggest reminding readers that dogs completed one 30-minute session, and the entire visit lasted about an hour when you include pre-session and post-session procedures (if accurate). This places the total time well-within the 2 hour limit.
Line 210, Section 2.4. Physiological Measurements and Behavior Evaluation: Consider adding a statement at the end of each sub sub section about what direction (lower or higher) would be associated with a negative welfare outcome post-session. For example, at the end of the sub-sub section Cardiac activity, state that higher heart rate and lower heart rate variability post-session would indicate stress. And after the sub-sub section on Salivary cortisol and oxytocin indicate that higher cortisol and lower oxytocin would indicate stress, and do the same for Tympanic membrane temperature. Not all readers may know these patterns.
Line 262: As mentioned above, it would help readers to better understand your study and put your behavioral results in context if you provide the FAS scale as an appendix or at least describe it further in the text. All the readers see in the Results section is that the before and after values (0.48 and 0.46) in Table 2 are similar, but how were these scores obtained? More detail/background is needed.
Results:
Lines 302-305: This text is not a complete sentence and seems out of place in the Dog Behavioral Evaluation section.
Discussion:
Lines 345-349: Clarify that “passively involved” in AAT in your previous study means not directly involved in the treatment program, if accurate. Does this mean the dogs and their handlers just visited the patients with fibromyalgia?
Conclusions:
Lines 368-371: The fact that this is the first such study should also be stated in the last paragraph of the Introduction to explicitly state what is new about your study, and possibly in the Abstract as well.
Author Response
|
This study examined responses of therapy dogs to participation in an animal-assisted rehabilitation program for stroke patients. Strengths of this study include a nice sample size for this type of work (14 pairs of therapy dogs and their handlers), multiple physiological measures plus behavioral measures used to assess welfare, and its design as a prospective randomized-controlled study. One weakness was failure to include the Canine Fear, Anxiety, and Stress scale as an appendix and/or provide more descriptive information about this scale so that readers could put the behavioral results in a larger context. Most of my comments are editorial in nature and are detailed below. |
Thank you very much for reviewing this manuscript. More description about the Canine Fear, Anxiety, and Stress scale has been provided. Below, please find our responses to your comments. |
|
Abstract: If possible, I would include that the study was prospective in nature and randomized-controlled somewhere early in the Abstract. |
The following sentence was added in the abstract. “This is the first randomized-controlled and prospective study that investigated the wellbeing of therapy dogs in an inpatient stroke rehabilitation program.” |
|
Introduction: Lines 57-58: Given the numerous terms used when describing human - nonhuman animal interactions, it might be helpful to differentiate AAT and AAI, mentioning the latter is a broader term that encompasses AAT as well as other animal-assisted activities. |
The sentence has been edited. “Glenk (2017) published an article regarding ethical and welfare concerns related to the use of dogs in animal assisted interventions (AAI), an umbrella term encompassing AAT in which animals are included for human benefits through different practices [3].” |
|
Materials and Methods: Line 129: Why is it important to mention that the dogs were not fed a raw diet? |
Feeding raw diet increases the risk of zoonotic diseases transmission, such as Salmonella. Since therapy dogs can interact with immune-compromised patients, i.e., elderly patients or patients receiving chemotherapy, it is vital to minimize the risk. |
|
Table 1: Consider using the following abbreviations for dog sex and spay/neuter status: SF (spayed female), IF (intact female), NM (neutered male), and IM (intact male) and F and M for female and male handlers, respectively. This might improve column spacing and make the table more readable. These could be described in a footnote. Note that there is a footnote numbered 2 after the table stating D2 and D3 were littermates, but there is no 2 within the table. |
Abbreviations for sex were added. Footnote numbered 2 was removed. |
|
Line 207: I suggest reminding readers that dogs completed one 30-minute session, and the entire visit lasted about an hour when you include pre-session and post-session procedures (if accurate). This places the total time well-within the 2 hour limit. |
The following sentence was added. “In this study, dogs participated in a 30-minute session, with the entire process lasting approximately an hour for each visit, including pre-session and post-session procedures.” |
|
Line 210, Section 2.4. Physiological Measurements and Behavior Evaluation: Consider adding a statement at the end of each sub sub section about what direction (lower or higher) would be associated with a negative welfare outcome post-session. For example, at the end of the sub-sub section Cardiac activity, state that higher heart rate and lower heart rate variability post-session would indicate stress. And after the sub-sub section on Salivary cortisol and oxytocin indicate that higher cortisol and lower oxytocin would indicate stress, and do the same for Tympanic membrane temperature. Not all readers may know these patterns. |
Explanations of each physiological measurement and behavior evaluation have been added. |
|
Line 262: As mentioned above, it would help readers to better understand your study and put your behavioral results in context if you provide the FAS scale as an appendix or at least describe it further in the text. All the readers see in the Results section is that the before and after values (0.48 and 0.46) in Table 2 are similar, but how were these scores obtained? More detail/background is needed. |
More explanations about canine FAS scale have been provided.
“The Canine Fear, Anxiety, and Stress (FAS) scale is a tool that has been utilized in veterinary clinics and research to assess the stress level in dogs. Based on the observation of the behavior and body language (from panting, ears pinned back and tail tucked to barking, growling and snapping), this Likert-type scale (relaxed/severe stress) provides a global FAS score for each dog. A higher FAS score indicates that the dog is showing more pronounced signs of fear, anxiety and stress. For instance, a dog with a FAS score 0 is relaxed with loose facial muscles and body tension, whereas a dog scored 5 on the FAS scale is severely fearful, anxious and stressed, showing signs of tense facial and body tension, offensive or defensive aggression [9–11]. The FAS scale was administered immediately before and after the therapy session outside of the patient’s room.” |
|
Results: Lines 302-305: This text is not a complete sentence and seems out of place in the Dog Behavioral Evaluation section. |
The sentences have been edited.
“The Canine Fear, Anxiety, and Stress (FAS) scale is a tool that has been utilized in veterinary clinics and research to assess the stress level in dogs. Based on the observation of the behavior and body language (from panting, ears pinned back and tail tucked to barking, growling and snapping), this Likert-type scale (relaxed/severe stress) provides a global FAS score for each dog. A higher FAS score indicates that the dog is showing more pronounced signs of fear, anxiety and stress. For instance, a dog with a FAS score 0 is relaxed with loose facial muscles and body tension, whereas a dog scored 5 on the FAS scale is severely fearful, anxious and stressed, showing signs of tense facial and body tension, offensive or defensive aggression [9–11]. The FAS scale was administered immediately before and after the therapy session outside of the patient’s room.” |
|
Discussion: Lines 345-349: Clarify that “passively involved” in AAT in your previous study means not directly involved in the treatment program, if accurate. Does this mean the dogs and their handlers just visited the patients with fibromyalgia? |
The sentences have been edited.
“The increased right TMT post-session observed may be explained by the fact that the therapy dogs in this study were actively involved in AAT, meaning they participated in exercises alongside the patients, rather than a passive involvement where the dogs simply provide companionship without engaging in specific activities.” |
|
Conclusions: Lines 368-371: The fact that this is the first such study should also be stated in the last paragraph of the Introduction to explicitly state what is new about your study, and possibly in the Abstract as well. |
Emphasis about the prospective and randomized-controlled nature of this study has been added in the abstract and the last paragraph of the introduction. |
Reviewer 3 Report
Comments and Suggestions for Authors
This is a very well written article on a topic that is so relevant and needed to maintain optimal welfare for the dogs who work with their humans. I have a few thoughts/suggestions that I hope I can convey adequately. I have witnessed far too many dogs being subjected to situations and interactions when they are uncomfortable and the handlers do not always recognize that level of stress or distress. My comments are made in an effort to not jump to the conclusion that all (therapy) dogs love every interaction.
1. In early 2024, IAHAIO published and endorsed new terminology for AAI. The new recommended terms are Animal Assisted Services (in place of Interventions), Animal Assisted Treatment (in place of therapy), Animal Assisted Support Programs (in place of Activities) and Animal Assisted Education remains the same. In an effort for a unified field, I would recommend making that shift.
2. You mention that your study was Animal Assisted Therapy (Treatment). It would be helpful to add a sentence or two on the definition or why your study qualified as AATx; specifically, who conducted the therapy and whether you used the diamond model or triangle model.
3. As a dog behavior consultant and trainer for many, many years, it is my understanding that a veterinarian does not receive much training in the way of behavior and it seems that having a dog trainer or behaviorist conducting the assessment might have been helpful. I don't mean any disrespect, but there is a risk of bias as the lead researcher uses a somewhat subjective assessment in determining whether the dog is stressed. Sometimes signs of stress are subtle (eg whiskers forward, excess dander, excess saliva, dilated pupils etc) that do not show up on that scale.
4. The study you conducted appeared very clean and considering physiological as well as physical indicators of stress. My concern comes from the idea that saying that dogs are "less stressed" after interacting with humans in a stressful setting implies that the dog benefits from the interactions (which may be true). It also sets the stage for handlers to think that these visits are not stress inducing for dogs at all, which we all know, can change in every interaction. I am also wondering what the dogs' 'baseline is at home. I would imagine that their baseline scores at the hospital would already be elevated since leaving their house (whether eustress, stress, or distress). How much it was elevated from home would seem to be significant. Once they are out of the room or session and their scores decreased, did they decrease because they left the interaction or because they "enjoyed" the session. I think this is important to note within the paper. In other papers' on dog welfare in AAS, there are situations that have been shown to be more stressful on dogs (eg being on leash vs being off leash, being pet on top of the head vs on the neck/chest, novel visits vs familiar visits, etc.). While I think your study is excellent, I am very leery about promoting visits as 'stress alleviating' for dogs, if that makes sense.
Thank you for the opportunity to review this great work.
Author Response
|
This is a very well written article on a topic that is so relevant and needed to maintain optimal welfare for the dogs who work with their humans. I have a few thoughts/suggestions that I hope I can convey adequately. I have witnessed far too many dogs being subjected to situations and interactions when they are uncomfortable and the handlers do not always recognize that level of stress or distress. My comments are made in an effort to not jump to the conclusion that all (therapy) dogs love every interaction. |
Thank you very much for reviewing this manuscript. A paragraph about recognizing each visit as a potential source of stress, as well as closely monitoring canine response at home versus during AAT has been added. Below, please find our responses to your comments. |
|
1. In early 2024, IAHAIO published and endorsed new terminology for AAI. The new recommended terms are Animal Assisted Services (in place of Interventions), Animal Assisted Treatment (in place of therapy), Animal Assisted Support Programs (in place of Activities) and Animal Assisted Education remains the same. In an effort for a unified field, I would recommend making that shift. |
Thanks for pointing this out. We’ve changed these terms throughout the manuscript. In terms of the terminology of dog, since the two certified organizations in this study, Alliance of Therapy Dogs and Pet Partners, still use “therapy dog”, to avoid confusion, it might be best to stay with the term- therapy dog. |
|
2. You mention that your study was Animal Assisted Therapy (Treatment). It would be helpful to add a sentence or two on the definition or why your study qualified as AATx; specifically, who conducted the therapy and whether you used the diamond model or triangle model. |
The following sentence was added along with the reference.
“All AAT sessions were conducted using the diamond model, involving patients, therapy dogs, therapy dog handlers, and therapists.” |
|
3. As a dog behavior consultant and trainer for many, many years, it is my understanding that a veterinarian does not receive much training in the way of behavior and it seems that having a dog trainer or behaviorist conducting the assessment might have been helpful. I don’t mean any disrespect, but there is a risk of bias as the lead researcher uses a somewhat subjective assessment in determining whether the dog is stressed. Sometimes signs of stress are subtle (eg whiskers forward, excess dander, excess saliva, dilated pupils etc) that do not show up on that scale. |
Thank you for pointing this out. Yes, we did monitor those subtle signs as well. The researcher (HYS) who did the behavioral assessment is a Fear Free certified veterinarian and a certified dog trainer. He completed his PhD in human-dog behavioral interaction, and canine behavior assessment was a key part of his thesis, involving assessing behaviors of over 200 shelter dogs and over 4,000 minutes of canine behavior footages using ethograms. (Here is his publication profile: https://orcid.org/0000-0002-5630-8632)
As you mentioned, ideally, an inter-rater reliability test should be performed to minimize the subjective interpretation. However, due to patients’ privacy and the hospital logistic, this was not possible and was mentioned in the study limitation. Therefore, this study utilized other objective physiological approaches, and all data was interpreted as a whole to avoid any bias. |
|
4. The study you conducted appeared very clean and considering physiological as well as physical indicators of stress. My concern comes from the idea that saying that dogs are “less stressed” after interacting with humans in a stressful setting implies that the dog benefits from the interactions (which may be true). It also sets the stage for handlers to think that these visits are not stress inducing for dogs at all, which we all know, can change in every interaction. I am also wondering what the dogs’ ‘baseline is at home. I would imagine that their baseline scores at the hospital would already be elevated since leaving their house (whether eustress, stress, or distress). How much it was elevated from home would seem to be significant. Once they are out of the room or session and their scores decreased, did they decrease because they left the interaction or because they “enjoyed” the session. I think this is important to note within the paper. In other papers’ on dog welfare in AAS, there are situations that have been shown to be more stressful on dogs (eg being on leash vs being off leash, being pet on top of the head vs on the neck/chest, novel visits vs familiar visits, etc.). While I think your study is excellent, I am very leery about promoting visits as ‘stress alleviating’ for dogs, if that makes sense. |
Thank you for pointing this out. The following paragraph was added.
“Finally, while the pre- and post-session results indicated that the careful inclusion of therapy dogs in an inpatient stroke rehabilitation program causes minimal to no stress to the animals, it is important to remain mindful that each visit could still be a potential source of stress. Haubenhofer et al. reported that therapy dogs exhibited higher salivary cortisol concentrations on workdays compared to non-workdays . However, further in-vestigation is needed to determine whether the elevated cortisol levels are directly related to stress. Future research should account for therapy dogs’ baseline stress levels in the study design to provide more comprehensive insights.” |
Round 2
Reviewer 3 Report
Comments and Suggestions for Authors
Thank you for the revisions. With that additional clarification, I think this is an excellent publication.